# Evaluating how demography and temperature increase might alter the burden of congenital Toxoplasmosis in Africa

Fidisoa T. Rasambainarivo[1]*, Ingrid G. Nilsson[2], Devin E. Cheeks[2], Wenchang Yang[3], C. Jessica E. Metcalf[2]

1 Department of Biology, East Carolina University, Greenville, North Carolina, United States of America, 2 Department of Ecology and Evolutionary Biology, Princeton University, Princeton, New Jersey, United States of America, 3 Department of Geosciences, Princeton University, Princeton, New Jersey, United States of America

* rasambainarivof23@ecu.edu

## Abstract

The impact of climate change on environmental pathogens is a question whose importance will amplify in coming years. The protozoan parasite and global zoonosis *Toxoplasma gondii* is one such: empirical evidence indicates that oocyst survival is reduced at high temperatures. Paradoxically, a decline in incidence of *T. gondii* infections could amplify the burden of this disease, as the most damaging outcome occurs subsequent to first infection during pregnancy, and reductions in the incidence of the infection will increase the average age of first infection. We blend models of infection dynamics rooted in occurrence across the African continent with models of human demography to bound expectations for the future burden of this pathogen, accounting for the effects of changing temperatures. We discuss targeting efforts and approaches for mitigation.

### Author summary

The pathogen *Toxoplasma gondii* is associated with serious health outcomes if women are infected for the first time during pregnancy. In Sub-Saharan Africa, transmission is currently sufficiently high that most women are infected before their child-bearing years. However, higher temperatures might reduce survival of the pathogen's environmental life stage (oocysts). This could reduce transmission and put more women at risk. We evaluate the potential shift in burden associated with a changing climate in the context of changing patterns of fertility over age, identifying settings where an increase might be expected.

**Data availability statement:** All data used in this study is open access and available freely on the internet, see the methods section for more details. Data and code used to produce this analysis is available from a Github repository cited in the manuscript https://github.com/fidyras/project_toxo.

**Funding:** FTR, IGN, and CJEM were supported by the High Meadows Environmental Institute (HMEI), Princeton University. CJEM was also supported by the Princeton Catalysis Institute. FTR was additionally supported by the Saint Louis Zoo WildCare Institute. The funders had no role in study design, data collection and analysis, decision to publish, or preparation of the manuscript.

**Competing interests:** The authors have declared that no competing interests exist.

## Introduction

Toxoplasmosis, caused by the protozoan parasite, *Toxoplasma gondii*, is a worldwide zoonosis (Fig 1) and a leading cause of severe foodborne infection [1,2]. Infection poses risks to vulnerable populations, notably pregnant women as the parasite can be transmitted to the fetus causing congenital defects including occulitis, encephalopathy, fetal death and stillbirth, a set of outcomes termed 'congenital toxoplasmosis' [3]. It has been estimated that congenital toxoplasmosis leads to more than 800,000 disease-adjusted life years (DALYs)/year globally [1], particularly in South America and sub-Saharan Africa [4,5].

The large majority of cases of congenital toxoplasmosis occur in women experiencing their first infection during pregnancy [6], although cases in women following infection with a different serotype, or resurgence in immunosuppressed mothers have also been reported [7]. This type of risk profile, where the highest burden is concentrated in women experiencing a primary infection during pregnancy can result in counter-intuitive effects of changes in the force of infection, defined as the probability that susceptible individuals become infected. As described for rubella [8], if the force of infection is very high, most individuals will be infected for the first time early in life, and thus will not be at risk during pregnancy; if the force of infection is very low, very few individuals will be infected at all, and thus the burden will be low. Intermediate levels of the force of infection may thus convey the greatest burden within a population (Fig 2).

Age seroprevalence studies can be used to estimate the force of infection, revealing patterns of risk across different settings [9]. Age seroprevalence for *T. gondii* in human populations is likely to be context specific, as the life cycle of this protozoan pathogen is mediated by behavioral and social features, such as food preparation methods, but also the environment (Fig 1). Humans and other vertebrates can be infected with *T. gondii* via two different paths: i) ingestion of *T. gondii* cysts by consumption of an infected animal; and ii) ingestion of oocysts directly from the environment (e.g., via unwashed leafy vegetables). Oocysts shed in the feces of cats are an important source of infection for vertebrates including humans. Crucially, experimental evidence indicates that the survival of *T. gondii* oocysts in the environment is reduced at high temperatures [10]. Although processes in natural systems are unlikely to map perfectly to tightly controlled experiments, this experimental result, alongside evidence that oocyst survival is reduced on freezing [11] suggests that oocysts are sensitive to temperature, and thus might have performance limits at higher and lower levels of temperature that could modulate overall transmission of the pathogen. This latter point was our focus in this paper: the age specific manifestation of the main burden of this infection in humans means that the effects of climate change on oocyst survival might result in large shifts in the impact on population health by reducing environmental exposure (Fig 1, red arrows) by diminishing survival of oocysts, which will reduce the rate at which susceptible individuals become infected, or the force of infection (Fig 2). However, other dependencies including on local demography mean that the mapping is unlikely to be straightforward.

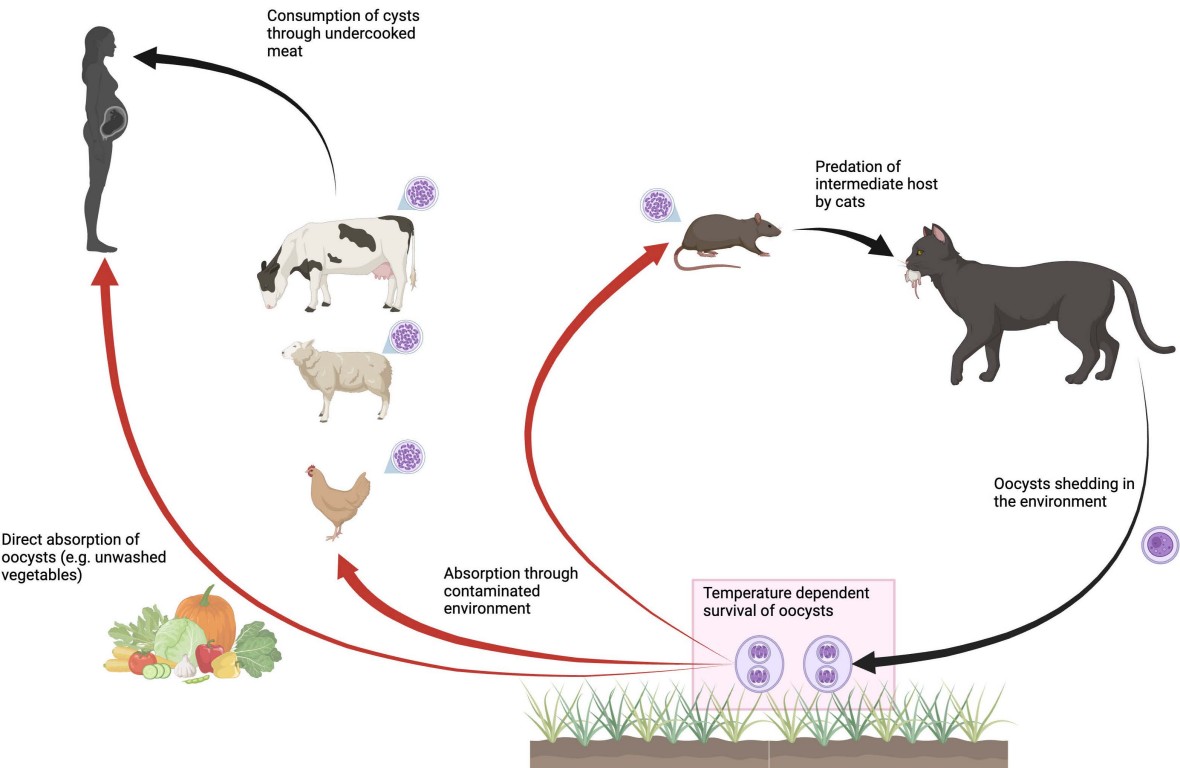

**Fig 1. Life cycle of T. gondii: Members of the Felidae family (including domestic and wild cats) are the only definitive hosts of T. gondii.** Cats and wild felids are infected by consuming the tissues of an infected intermediate host (almost any vertebrate species), that has itself become infected typically by ingesting oocysts in the environment and thus having developed *T. gondii* cysts (purple). Cats may also be infected via congenital transmission. Oocysts are then shed in the environment through the cats' feces and sporulate 1 to 5 days later, becoming infective to other intermediate or definitive hosts. Humans and other intermediate hosts can also become infected by consuming food or water contaminated with oocysts as well as the consumption of infected intermediate hosts and thus infection via absorption of *T. gondii* cysts. High temperatures reduce the survival of oocysts in the environment, and this might thus reduce rates of infection of intermediate hosts (red arrows indicate temperature dependent rates), with knock-on effects across the food chain. Created in BioRender. Rasambainarivo, F. (2026). https://BioRender.com/9tb3yvo.

Here, we combine seroprevalence studies across Africa to estimate the force of infection of this pathogen in different settings through time, and estimate how this is modified by temperature and the human development index, used as a proxy for a suite of hygiene features likely to reduce the risk of toxoplasmosis infection. These estimates of the force of infection and its dependencies allow us to estimate the risk of primary infection across age and combine this with current and future estimates of fertility to evaluate the changing burden of congenital toxoplasmosis, comparing the effects of demographic change (specifically, age structured and fertility profiles in 2023, 2050 and 2100), and temperature change, here evaluating the extreme scenario of a three degree increase in temperature. Our analysis provides a framework to anticipate future disease burden and to identify priority regions for targeted interventions under climate and demographic change.

## Methods

### Serology data

Seroprevalence studies categorize individuals as seronegative, i.e., having never been infected by *T. gondii* ($Y_i = 0$), and seropositive, i.e., chronically infected ($Y_i = 1$). Totals by age class are reported across the literature. We leveraged

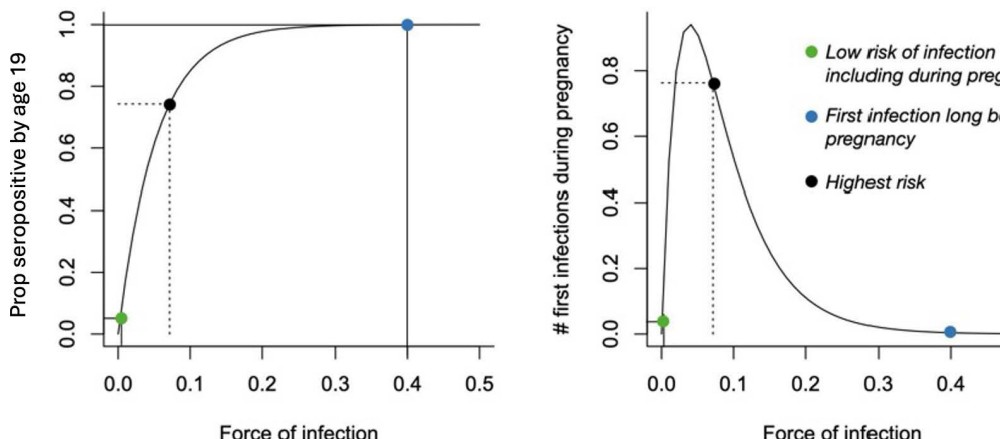

**Fig 2. Schematic indicating the paradoxical effect relating the force of infection (FOI) and burden of congenital toxoplasmosis, assuming for simplicity that pregnancies start occurring in individuals aged > 19 years.** As the force of infection increases (x axis, left hand side), the proportion of individuals seropositive by age 19 (y axis, left hand side) will also increase. For individuals living in a location with a high force of infection (blue) almost everyone will already have been infected by age 19 (left hand side, blue point at ~100% seropositive), which will mean that very few individuals will be at risk of acquiring a new infection during pregnancy (right hand side, blue point close to zero). Conversely for individuals living in location with a very low force of infection, very few individuals will be infected at all (left hand side, green point close to zero), so although many individuals remain susceptible by age 19, they are at little risk of acquiring the infection, and thus the risk of a first infection occurring during pregnancy is also low (right hand side, green point close to zero). Individuals living in settings with an intermediate force of infection (black) are potentially at the highest risk, as they are neither guaranteed to have already been infected (blue point), nor at very low risk (green point); thus, paradoxically, in high FOI settings, a decline in the FOI can result in an increase in burden depending on details of distribution of fertility.

a meta-analysis of seroprevalence across pregnant women [9], where authors searched through various databases to retrieve pertinent findings about the prevalence of toxoplasmosis infection in Africa using the following Mesh terms ("Toxoplasmosis" OR "Toxoplasma gondii"), AND ("Pregnant Women" OR "Prenatal care"), AND ("Africa"), as well as combinations including the names of individual African countries. We further expanded the dataset by conducting citation-based snowballing, in which we examined the references of included papers and subsequent studies citing them. Eligible studies reported seroprevalence in people and provided sufficient information on age groups, sample sizes, and prevalence estimates by age. We included studies that used any diagnostic method for detecting *T. gondii* antibodies (S2 Fig), restricting the search to those conducted in Africa since 1979 when detailed temperature data became available. Data for each article were extracted using a previously prepared checklist including Name of first author, Year of publication, year of sampling, sample size, age range, number of seropositives (or percentage of seropositives) for each age class (S1 Table).

### Other covariates

Previous work indicates that climatic variables including temperature might alter pathogen ecology, with, in particular, lower survival of oocysts at higher temperatures [10] S1 Fig presents a statistical analysis of the effect of temperature on oocyst survival based on data taken from this paper. For each study available, we therefore obtained minimum and maximum temperature using the nearest urban center as described and using the daily ground temperature over the years 1979–2024 from the fifth generation European Centre for Medium-Range Weather Forecasts atmospheric reanalysis (ERA5) of the global climate [12] (S3 Fig). ERA5 combines both in-situ/satellite observations and weather forecast model simulation information to create the best estimate of global-coverage climate information on regular longitude/latitude grids (with an approximate horizontal resolution of 30km). It is a popular dataset used in climate studies and often treated as "observations" when compared to climate model simulations. Daily mean ground temperatures at locations of interest in this study, given their information of longitude and latitude, are obtained by bilinear interpolation from the globally gridded

data. Since biological limits across abiotic variables like temperature are expected, we included both annual minimum and maximum of the daily mean temperatures. The variance inflation factor (calculated using the R package rvif) indicated that multicollinearity between these two variables was sufficiently small as to not destabilize the regression (V = 1.08) as indicated in S3 Fig. For serology data preceding the first available temperatures measurements [Mauritania, 1973] we took values representing 1979. Excluding these measurements did not qualitatively alter results.

The complex ecology of this pathogen (Fig 1) indicates that different patterns of sanitation and cultural practices (food preparation) can modulate the risk of human infection. Addressing this in detail would require much finer anthropological and social data than currently available; however, larger scale proxies may capture some of the key details. For each country/year combination, we extracted the Human Development Index from OurWorldInData, a metric that blends education, health and economic indexes [13]. To capture additional cultural and other contextual variation, we fit 'country' as a random effect in what follows.

## Estimating the force of infection

Data indicating seroprevalence by age opens the way to estimating the force of infection, or rate at which susceptible individuals become infected, defined as $\lambda$. In the simplest analysis, the probability of evading infection up until time of sampling t is $S(t) = exp(-\lambda t)$, and the probability of having become infected by time $t$ is the complement of this $F(t) = 1 - exp(-\lambda t)$. It follows that $Y_i \sim Bernouilli(F(t_i))$ and we can thus use a generalized linear model framework to estimate the force of infection [14]. To formally account for covariates modulating the force of infection, we use the complementary log-log link:

$$g(F(t_i)) = log(-log(1 - F(t_i))) = \alpha + log(t_i)$$

where $\alpha = log(\lambda)$; a relationship that can be fit using the glm function in R, setting log age as an offset. To reflect the fact that the force of infection might be modulated by temperature [10] as well as social context features, we assume that

$$\lambda = exp(\beta_0 + \beta_1 T_{min} + \beta_2 T_{max} + \beta_3 HDI + c)$$

where $\beta_0$ is an overall intercept, $\beta_1$ and $\beta_2$ are slopes on minimum and maximum annual temperatures respectively, $\beta_3$ is a slope on the human development index and $c$ is drawn from a distribution centered on zero capturing the range of country specific effects. We fit this model using the lmer package in R where the full link function is defined by:

$$g(F(t_i)) = \beta_0 + \beta_1 T_{min} + \beta_2 T_{max} + \beta_3 HDI + c + log(t_i)$$

and obtain estimates and standard errors of the full set of parameters.

## Impact on burden of congenital toxoplasmosis

Increasing the force of infection, $\lambda$, or the rate at which susceptible individuals become infected reduces the average age of first infection, $A$, with $A = 1/\lambda$. Changes in global demography shift the age at which women are having children, via both changes in the age distribution of the population, and changes in the age at which women choose to have children. The burden of congenital toxoplasmosis will be determined by the intersection between these two processes (schematically depicted in Fig 2), with details of the magnitude of effects contingent on the details of the associated distributions. To capture this, we use estimates and forecasts provided by the UN Population Division, focussing on 'Births by Age of Mother' [15], which accounts for both shifts in age structure, and changing patterns of fertility over age (see United Nations, Department of Economic and Social Affairs, Population Division (2024) for a discussion of how uncertainty is evaluated in these projections). We do not provide estimates of absolute burden, but rather focus on the relative burden,

*r* (reflecting now vs. future scenarios), in order to broadly generalize across the nuances of timing of infection over the course of pregnancy relative to circulating serotypes, differential virulence of different strains in different regions [7], etc. The probability that an individual is infected at age *a* is defined by:

$$P_a = \lambda exp(-\lambda a)$$

where the second term $exp(-\lambda a)$ captures the probability of remaining susceptible up to age a, and the first term the risk of infection at that age. Thus, the relative burden for each country is defined by:

$$r = \frac{\lambda_f exp(-\lambda_f a)\ B_{f,a}}{\lambda_n exp(-\lambda_n a)\ B_{n,a}}$$

where $B_{f,a}$ is the future number of children born to women of age *a* (either in 2050 or 2100, using the 'future medium variant' provided by the UN Population Division) and $B_{n,a}$ is the current (or most recent estimate, i.e., 2023) number of children born to women of age *a*; and $\lambda_f$ is the predicted future hazard of infection, $\lambda_n$ is the current hazard of infection, based on the most recent estimates of HDI (2022) and temperature (2024).

For $\lambda_f$, we explore scenarios where the future climate remains constant (such that only the demography changes via $B_{f,a}$ and $B_{n,a}$, and we retain minimum and maximum temperatures reflecting 2024), or that under future climates, the minimum 2024 temperature is retained, but the maximum temperature increases by 3 degrees celsius, vice versa, or both. In all scenarios, to isolate the effects of climate and demography, we assume that the country-specific HDI is constant at levels reported for 2022.

## Results

A total of 92 articles documenting the seroprevalence of *Toxoplasma gondii* in 80,805 people from a total of 25 countries in years from 1979 to 2023 were reviewed and analyzed in this study (see PRISMA flow diagram, S2 Fig). The full range of the data is shown in Fig 3, and comprises 402 data-points across 25 countries.

The fitted model (S2 Table) explains 52% of the variation in seroprevalence and indicates that the force of infection declines with the HDI (Fig 4A) and maximum temperature (Fig 4B) and increases with minimum temperature (Fig 4C), also varying across countries. The width of age bins across the different studies is highly variable; to verify that our results were not sensitive to this, we estimated relationships using both the midpoint of age bins, and the upper age of age bins; finding largely consistent results.

Combining these results with estimates of current and future births across age ("World Population Prospects," n.d.-b), assuming a constant HDI (implying broadly invariant sanitation and dietary features in populations), and temperatures either remaining constant or increasing by 3 degrees celsius (the upper limit of expectations), we find a vast diversity of outcomes, ranging from halving to near doubling of the burden of congenital toxoplasmosis (Fig 5). Demographic effects largely dominate (the distance between the filled and hollow black points is greater than their distance to coloured points, Fig 5). An increase in maximum temperature in general decreases the burden of congenital toxoplasmosis, while effects of increases in the minimum temperature or both minimum and maximum temperature are variable. The direction of effects resulting from demographic changes are variable: some countries experience an increase in burden (e.g., Tanzania, Ivory Coast) and some a decrease (e.g., Morocco, Tunisia).

To unpack this complexity, we fitted a linear model to log projected burden, with explanatory variables including the type of temperature change (degree increase in the maximum or minimum temperature, exploring values of 0, 1, 2 and 3), the year of demographic projections (current, 2050, 2100), the average country HDI, and the interaction between these latter two, leveraging covariation between demographic patterns and HDI to unpack the effect of different time-horizons of demographic change. Fitted estimates (S3 Table) indicate that, on average, increases in the maximum temperature

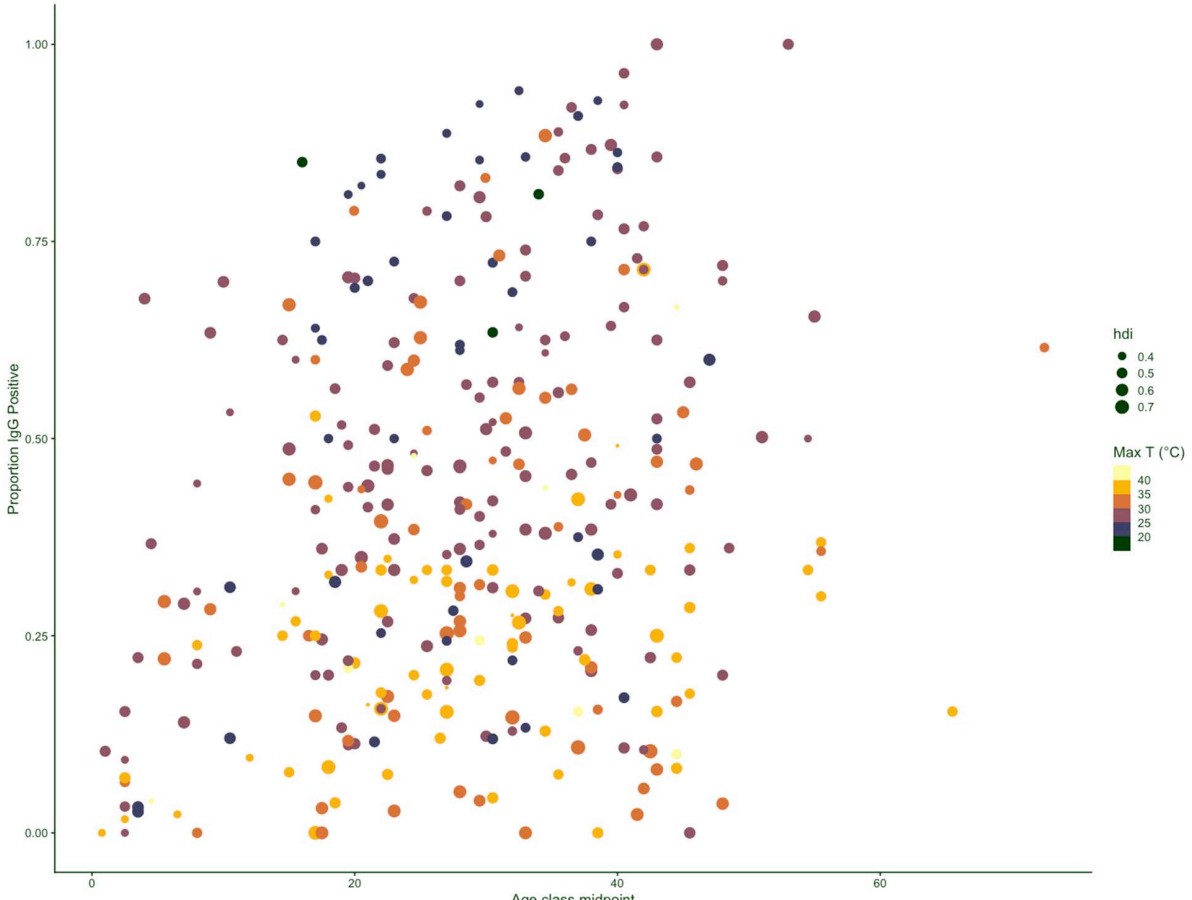

**Fig 3. Seroprevalence data extracted from the literature.** Each data point indicates a combination of age class midpoint (x axis) and proportion seropositive (y axis). Points are coloured by maximum city temperature at time of study. Dot size is proportional to the Human Development Index (HDI) of the country at the time of the study. The concentration of warmer colours (indicative of higher temperatures) towards the bottom of the plot suggests a reduction in transmission associated with high temperatures as suggested by 8.

reduce the relative burden; increases in the minimum temperature increase it; and the relative burden increases with year more in countries with a higher HDI, suggesting that their demographic structure translates into a concentration of risk in women of childbearing age in the future. Lower HDI countries may thus reflect demographic scenarios where both high force of infection and early age of fertility combine to keep risk low even in future potentially warmer climates.

## Discussion

Global change will modulate the future burden of toxoplasmosis via multiple channels [16], including changes in cultural and sanitation practices, changes in temperature affecting pathogen survival, and demographic changes in human populations. Here, we leveraged existing age-serology patterns from across the African continent to provide a preliminary set of estimates of how the broad human context (evaluated via the human development index) and maximum and minimum temperature modulate risk of infection. Our results align with expectations: risk declines with HDI and maximum temperature - and also enable us to explore future burden under projections of demographic change across a range of temperature changes. However, these results should be interpreted as an initial step towards integrating demographic, climatic and epidemiological processes, and there remains substantial "*avenues*" for refinement.

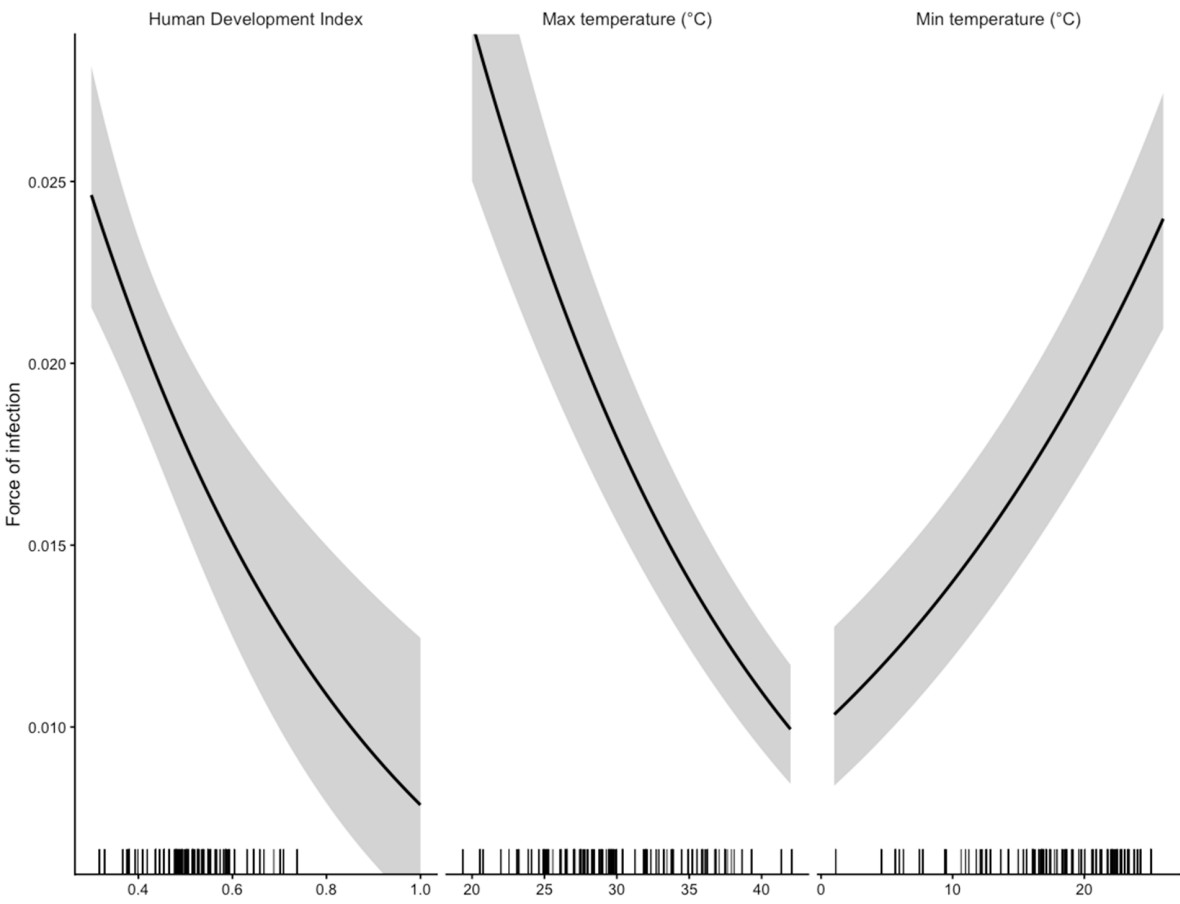

**Fig 4. Fitted estimates for the force of infection (y axis) with mean (solid line) and standard error (dashed line) across variables on the x axis; other variables are set to their average within the data-set.** Tickmarks on the x axis indicate the location of the data. Results indicate (A) declines with the human development index (HDI, x axis), a proxy for changes in access to sanitation, etc; (B) declines with maximum temperature (x axis), in line with estimated negative effects of increased temperature on oocyst survival (see S1 Fig); (C) increases with minimum temperature (x axis), suggesting a lower bound on oocyst survival, as observed in many biological systems. The full model explains 52% of the deviance.

A key component of our projections relies on demographic estimates provided by the United Nations Population Division. These data integrate observed census with model-based corrections to account for incomplete or biased reporting. As a result, our estimates should be interpreted as scenario-based projections rather than precise forecasts.

Moreover, our statistical analysis has additional limitations. We did not fit variation in risk of infection with age, as the data would not allow it. Although a focus on minimum and maximum temperature may have provided us with a relatively clear signal, as the edges of the distribution of the biological range may be where the effects are most clear the true relationship with climatic variables is likely to be more nuanced (e.g., modulation via median temperature, or effects of rainfall [17]). Further experimental analyses of *T. gondii* oocyst responses to temperature alongside further observational studies, perhaps across targeted temperature gradients could help to address this. However it is also important to recognize that increases in temperature of the scale explored here are likely to lead to much wider ecosystem disruption, for which impacts are likely to be hard to predict. We also did not have data available to address potentially different burdens associated with variation across infecting genotypes in severity and burden of CT [18], or risk associated with infection in mothers immune to a typical genotype challenged by an atypical genotype during pregnancy [19,20]; although our focus

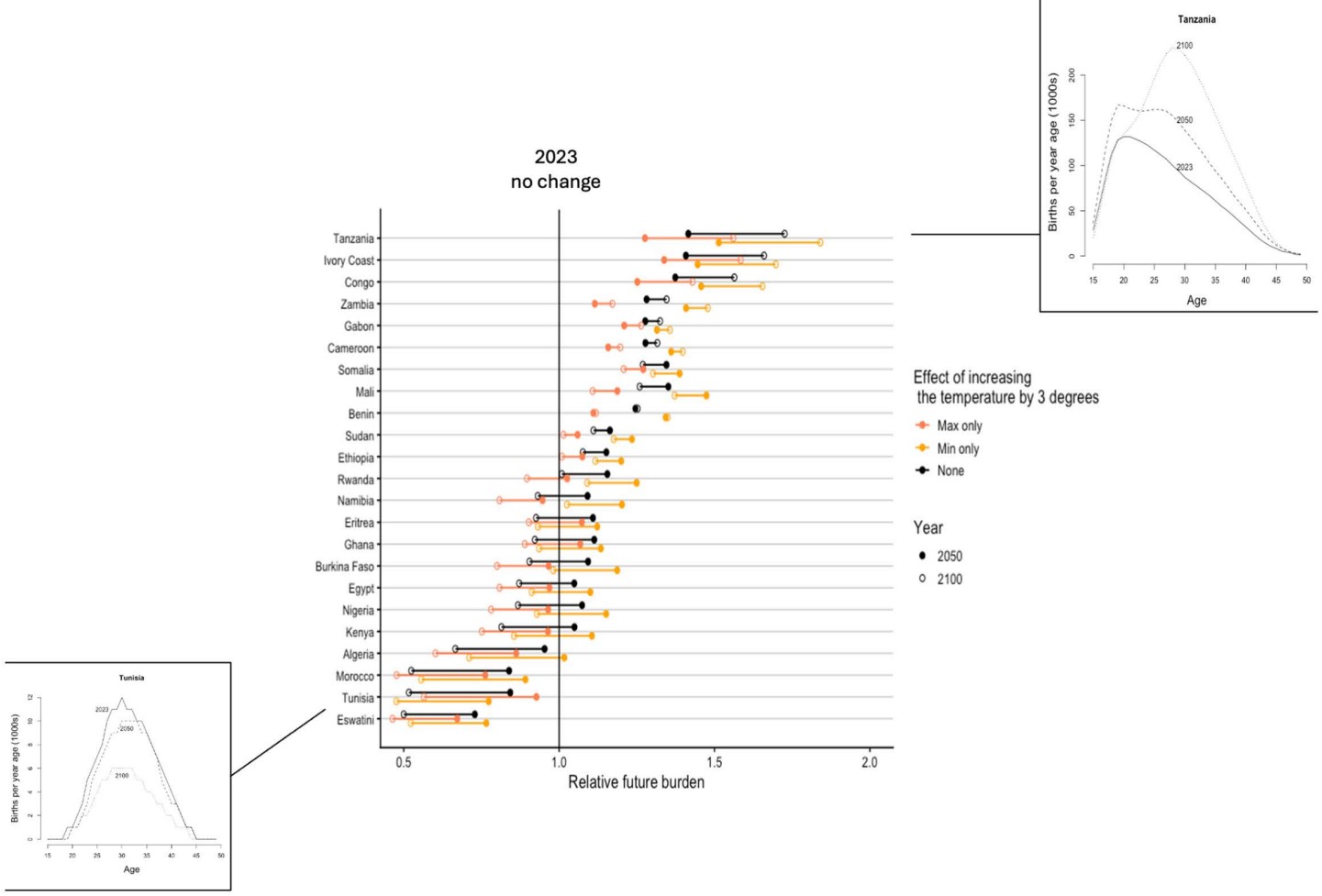

**Fig 5. Relative future burden calculated as the total number of children born to a mother experiencing their first infection during pregnancy and thus at risk of congenital toxoplasmosis in 2050 (filled points) and 2100 (hollow points) relative to 2023 (vertical line, x = 1), indicating effects at contemporary temperatures (black), and under increases of 3 degrees of the maximum (coral), or minimum (yellow) temperature (a range of intermediate values are shown in S5 Fig).** Two countries that are examples of extreme relative burden effects are highlighted, Tanzania and Tunisia. In the former, burden is increasing as a result of increasing total numbers of births (lines representing births in 2100 exceed 2050 which exceed 2023); while the opposite is observed in Tunisia. Controlling for the effect of total numbers by estimating the change in proportion of births instead indicates that relative proportion burden declines in countries where the age at reproduction is increasing (as most mothers will then be infected before pregnancy) and these countries also show smaller relative impacts of temperature change (S6 Fig). Note that no fertility estimates are available for São Tomé and Príncipe.

on relative country comparisons should control for this as long as there are no major future changes in global genotype circulation, or infections associated with travel or transport of consumables. Projections also neglect the role of other potentially important drivers of change, such as an increase in consumption of meat, or improvements in hygiene (noting that the two might counterweight each other). Nevertheless it provides a first foundation for considering the future burden of this global and important pathogen.

Temporal declines in *Toxoplasma gondii* seroprevalence have been observed in several human populations, potentially indicating a reduction in congenital toxoplasmosis (CT) incidence in some settings [21]. The drivers of this decline are likely multifactorial and may include improvements in sanitation and hygiene, better food safety practices, and cat

population management. However, our synthesis of projected burdens under future demographic and climatic scenarios suggests a complex, and at times counterintuitive, risk landscape for congenital toxoplasmosis.

This dynamic set of possible outcomes for CT parallels projections for congenital rubella syndrome (CRS) associated with introduction of vaccination [22,23]. Rubella is a directly transmitted completely immunizing infection, and pre-vaccination, most rubella infections occur during childhood, and are relatively mild. Introduction of the vaccine at scales insufficient to achieve herd immunity, has the potential to slow down acquisition of immunity and shift susceptibility into reproductive ages, increasing incidence of CRS. This 'paradoxical' outcome where partial control measures can inadvertently heighten the risk to fetuses is referred to as a peak shift dynamic [24,25]. A similar dynamic could emerge for CT: without appropriate adaptation of public health strategies, declining *T. gondii* exposure may increase susceptibility among women of reproductive age, particularly in populations undergoing rapid urbanization or socioeconomic transition, or rapid increases in temperature. Anticipating this shift will require proactive targeting of interventions. In particular, screening and education programs for at-risk populations (women of childbearing age) should be prioritized in regions where burden is projected to rise. At present, prenatal serological screening remains the only diagnostic tool capable of identifying all potential cases of congenital toxoplasmosis (CT) in time to provide appropriate diagnosis, treatment, and follow-up ([26]; "Prevention and Mitigation of Congenital Toxoplasmosis. Economic Costs and Benefits in Diverse Settings" 2019 [27]). This approach should be complemented by strengthening and ensuring access to diagnostic capabilities and potentially prophylactic measures in at-risk regions.

While no human vaccine is yet available, development efforts are ongoing [28,29] and the eventual deployment of any vaccine would need to be guided by robust cost–benefit analyses to avoid unintended epidemiological consequences. Lessons from other systems highlight the importance of these economic considerations as they may shape the uptake and effectiveness in a future human *T. gondii* vaccination program. Although the development of a human vaccine against *T. gondii* would be transformative globally, its realistic implementation in Africa will also depend on sustained investment in research capacity, biotechnology infrastructure, and health-system strengthening. A One Health approach that coordinates across human, animal, and environmental health sectors will be critical to manage transmission risk holistically [30,28,31]. Such strategies could include targeted vaccination of cats to reduce environmental contamination by limiting oocyst shedding [32].

Finally, this work contributes to a growing body of evidence that climate change does not uniformly reduce or exacerbate disease risks, but instead reshapes the distribution and nature of those risks across populations, and that future human demography remains a key determinant of disease burden.

## Supporting information

**S1 Fig. Effect of temperature on oocyst survival.** [10] evaluated success or failure of experimental infection of mice following oocyst exposure to different temperatures (35, 40, 45, 50 and 55 degrees for variable durations; n = 62, with 35 events. We digitized the data, and used survival analysis to explore the effects of temperature on oocyst survival. Results from this Kaplan Meier analysis estimates are depicted here and indicate that above 40 degrees celsius, oocyst survival (measured as ability to infect susceptible mice) is considerably reduced. The data and code are available from https://github.com/fidyras/project_toxo.
(TIFF)

**S2 Fig. Selection process of studies according to the PRISMA flow diagram.** This study was conducted following the 2020 Preferred Reporting Items for Systematic reviews and Meta-Analyses (PRISMA) criteria. Initial literature search was conducted in eight databases using keywords in titles, e.g., "toxoplasmosis seroprevalence africa". Collected papers were recorded in an excel spreadsheet, as well as saved in PDF form in a OneDrive folder. Duplicate papers were merged. Secondary screenings of abstracts were applied based on inclusion and exclusion criteria, and four papers were excluded due to study not meeting

inclusion criteria. The remaining papers were reviewed to see if they matched eligibility criteria. Among excluded criteria, two texts were found to be incomplete, two were with non target populations (e.g., HIV positive individuals), three included a location scope that was too large (e.g., a whole country or region's data, nonspecific GPS points to view surface temperature), one study was too old (e.g., collected before ERA data began in 1979), and six studies were not categorized by age group and thus ineligible for inclusion. Finally, remaining eligible papers were included and recorded for data analysis in an excel spreadsheet. In search of more papers, the snowball method was used on eligible papers. This method involved screening article titles in the reference section of eligible papers for key phrases such as "toxoplasmosis seroprevalence". Adding papers by this method allows us to diversify the collection of papers, including to more countries and larger temporal breadth. Articles collected this way underwent the same rigorous screening and are included in the numbers reflected above. In total, 92 papers were determined eligible for study analysis, 72 of which were studies done on pregnant women and women of reproductive age. (TIFF)

**S3 Fig. Plot indicating minimum and maximum temperature values corresponding to locations and years for which seroprevalence data was available.** See data for details.
(TIFF)

**S4 Fig. Country-level random effects from the fitted statistical model. C**ountry level effects from the fitted statistical model depicted in Fig 4, capturing residual variation associated with country to country variation. See S2 Table for the full set of parameter estimates.
(TIFF)

**S5 Fig. Projected relative burden under temperature and demographic change scenarios.** Projected relative burden of Toxoplasma gondii infection under scenarios of no temperature change and increases of +1°C, +2°C, and +3°C in maximum (x-axis) and minimum (y-axis) temperatures for the years 2023, 2050, and 2100. Each panel represents a temperature-change scenario, and projections incorporate corresponding demographic changes for each time period. Colour scales are standardized across panels, with deeper red indicating a higher relative burden and lighter colours indicating a lower relative burden. Future demography has larger effects in countries with ongoing population growth.
(PDF)

**S6 Fig. Relative future proportional burden calculated as the proportion of children born to a mother experiencing their first infection during pregnancy and thus at risk of congenital toxoplasmosis in 2050 (filled points) and 2100 (hollow points) relative to 2023, indicating effects at contemporary temperatures (black), and under increases of 3 degrees of the maximum (orange), or minimum (coral) temperature (S6A Fig).** The right hand panel (S6B Fig) indicates the relative future burden in proportion of CT births in 2100 compared to 2023 against the change in mean mother's age for children born calculated by subtracting values for 2023 from 2100, indicating that as the age at fertility increases (a function of both trends in choice and a shifting population structure) the proportional burden of CT will decline, since most mothers are then likely to be infected prior to pregnancy. Effects of temperature emerge as larger where changes in age of fertility are smallest.
(TIFF)

**S1 Table. Summary of the seroprevalence studies of Toxoplasmosis across Africa** included in the review.
(DOCX)

**S2 Table. Parameter estimates for the model fitted to force of infection (Fig 4) are: an intercept of -2.284 (standard error 0.348), a coefficient on HDI of -1.500 (0.351), a coefficient on maximum temperature of -0.067 (0.003) and a coefficient on minimum temperature of 0.060 (0.004); country specific estimates are provided below.**
(DOCX)

**S3 Table. Parameters for a model synthesizing effects on log projected relative burden.** Projected burden was obtained by estimating the force of infection for each country assuming that HDI was unchanged, but maximum and minimum temperature increased by 0–3 degrees celsius (resulting in 16 possible combinations of minimum and maximum). The resulting age profile of first infection was combined with projected number of children born to women of age *a* in the focal year (2023, 2050 and 2100). Relative burden was obtained by standardizing by dividing each value by estimates obtained for that country in 2023 with no temperature change. We fitted minimum and maximum temperature as quantitative variables, alongside year and HDI, and an interaction between the latter two ($r^2 = 0.25$).
(DOCX)

## Author contributions

**Conceptualization:** Fidisoa T. Rasambainarivo, C. Jessica E. Metcalf.

**Data curation:** Fidisoa T. Rasambainarivo, Ingrid G Nilsson, Devin E Cheeks, Wenchang Yang, C. Jessica E. Metcalf.

**Formal analysis:** Fidisoa T. Rasambainarivo, Ingrid G Nilsson, Devin E Cheeks, Wenchang Yang, C. Jessica E. Metcalf.

**Investigation:** Fidisoa T. Rasambainarivo, Wenchang Yang, C. Jessica E. Metcalf.

**Methodology:** Fidisoa T. Rasambainarivo, Ingrid G Nilsson, Devin E Cheeks, Wenchang Yang, C. Jessica E. Metcalf.

**Project administration:** Fidisoa T. Rasambainarivo, C. Jessica E. Metcalf.

**Resources:** C. Jessica E. Metcalf.

**Software:** Wenchang Yang, C. Jessica E. Metcalf.

**Supervision:** C. Jessica E. Metcalf.

**Visualization:** Fidisoa T. Rasambainarivo, Ingrid G Nilsson, Devin E Cheeks, Wenchang Yang, C. Jessica E. Metcalf.

**Writing – original draft:** Fidisoa T. Rasambainarivo, Ingrid G Nilsson, Devin E Cheeks, Wenchang Yang, C. Jessica E. Metcalf.

**Writing – review & editing:** Fidisoa T. Rasambainarivo, Ingrid G Nilsson, Wenchang Yang, C. Jessica E. Metcalf.

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
