## [Decision Letter · Decision Letter 0]

12 Nov 2025

The future burden of congenital Toxoplasmosis in Africa under demographic and climate change

Dear Dr. Rasambainarivo,

Thank you for submitting your manuscript to PLOS Neglected Tropical Diseases. After careful consideration, we feel that it has merit but does not fully meet PLOS Neglected Tropical Diseases's publication criteria as it currently stands. Therefore, we invite you to submit a revised version of the manuscript that addresses the points raised during the review process.

Please submit your revised manuscript within by Jan 11 2026 11:59PM. If you will need more time than this to complete your revisions, please reply to this message or contact the journal office at plosntds@plos.org. Please include the following items when submitting your revised manuscript:

We look forward to receiving your revised manuscript.

Kind regards,

Naomichi Yamamoto, Ph.D.

Academic Editor

Laura-Isobel McCall

Section Editor

Shaden Kamhawi

co-Editor-in-Chief

Paul Brindley

co-Editor-in-Chief

**Additional Editor Comments :**

All three reviewers have generally provided favorable comments on the manuscript. Please incorporate their comments to improve the manuscript. I have also added my own comments below.

I agree with reviewer 2's comment that the figures are generally difficult to read and understand. Please revise them to make them easier to see.

I agree with reviewer 3, who pointed out that one limitation of this study is that it assumed a temperature rise of 3 degrees. It would be even better to perform a sensitivity test to see how sensitive the results are to temperature changes (e.g., 1, 2, and 3 degrees change).

I failed to understand the mechanism by which rising minimum temperatures increases the force of infection (Figure 4C). Please explain the hypothesized mechanism behind.

There may be collinearity between the minimum and maximum temperatures, which are incorporated as independent variables into the single statistical model (pages 9-10). Collinearity might make the regression model unstable, so it is recommended that the authors test for collinearity using VIF or Pearson r.

Related to the question above, I failed to understand why the model includes two very similar, potentially interrelated explanatory variables (i.e., maximum and minimum temperatures) that might be causing the collinearity issue. Wouldn't it be more reasonable to use either the average temperature, the maximum temperature, or the minimum temperature and build a separate model for each temperature index (i.e., a total of 3 separate regression models are built with 3 different temperature index) and compare the results across temperatures? Please explain (or refute) why two potentially interrelated variables have to be incorporated into a single regression model.

The provided Github link cannot be reached, so please check the address of the link.

**Journal Requirements:**

At this stage, the following Authors/Authors require contributions: Fidisoa T Rasambainarivo, Ingrid G Nilsson, Devin E Cheeks, Wenchang Yang, and C. Jessica E. Metcalf. Please ensure that the full contributions of each author are acknowledged in the "Add/Edit/Remove Authors" section of our submission form.

5) We have noticed that you have uploaded Supporting Information files, but you have not included a complete list of legends. Please add a full list of legends for your Supporting Information file (Table S1) after the references list.

6) We notice that your supplementary Figures are included in the manuscript file. Please remove them and upload them with the file type 'Supporting Information'. Please ensure that each Supporting Information file has a legend listed in the manuscript after the references list.

7) Some material included in your submission may be copyrighted. According to PLOSu2019s copyright policy, authors who use figures or other material (e.g., graphics, clipart, maps) from another author or copyright holder must demonstrate or obtain permission to publish this material under the Creative Commons Attribution 4.0 International (CC BY 4.0) License used by PLOS journals. Please closely review the details of PLOSu2019s copyright requirements here: PLOS Licenses and Copyright. If you need to request permissions from a copyright holder, you may use PLOS's Copyright Content Permission form.

Potential Copyright Issues:

i) Figure 1. Please confirm whether you drew the images / clip-art within the figure panels by hand. If you did not draw the images, please provide (a) a link to the source of the images or icons and their license / terms of use; or (b) written permission from the copyright holder to publish the images or icons under our CC BY 4.0 license. Alternatively, you may replace the images with open source alternatives. See these open source resources you may use to replace images / clip-art:

ii) Figure S2 . We noted that you stated in the figure legend "Dubey et al. [10] ." Please clarify whether the figure is obtained from a previously published article. If so, please include a direct link to source of the figure in the legend and a link to the terms of use/license information.

8) Thank you for stating "All data used in this study is open access and available freely on the internet, see the methods section for more details." Please amend your Data Availability Statement to include the full link(s) to the data.

9) You indicated that "Data and code used to produce this analysis is available from a Github repository cited in the manuscripthttps://github.com/fidyras/project_toxo." When we accessed the provided link, this message was displayed "This repository is empty." Please provide us with the link where the data can be found.

10) Please revise your current Competing Interest statement to the standard "The authors have declared that no competing interests exist."

**Reviewers' Comments:**

Reviewer's Responses to Questions

**Key Review Criteria Required for Acceptance?**

**Methods**

-Are the objectives of the study clearly articulated with a clear testable hypothesis stated?

-Is the study design appropriate to address the stated objectives?

-Is the population clearly described and appropriate for the hypothesis being tested?

-Is the sample size sufficient to ensure adequate power to address the hypothesis being tested?

-Were correct statistical analysis used to support conclusions?

-Are there concerns about ethical or regulatory requirements being met?

Reviewer #1: The authors attempt to estimate the future burden of congenital toxoplasmosis in Africa under scenarios of demographic and climate change. The hypothesis that climate change could reduce the force of infection and thereby increase disease burden is interesting. However, I have several concerns about the study design and interpretation.

1) Toxoplasma gondii is transmitted by at least two major routes — ingestion of oocysts and ingestion of tissue cysts — and ingestion of tissue cysts is generally considered the dominant route in many settings. The relative importance of these routes can vary by country and context. The manuscript should explicitly acknowledge both oocyst- and tissue cyst–mediated transmission.

2) The hypothesis that higher temperatures reduce oocyst survival is derived from experimental studies conducted in warm water baths, which may not reflect the environmental conditions oocysts actually experience in the field under climate change. Although the fitted model shows a negative association between Tmax and the force of infection, the model explains only ~52% of the variation. To strengthen the argument, I recommend visually comparing the observed seroprevalence data with the temperature distributions (e.g., Tmin/Tmax ranges) across countries or study sites so readers can better assess whether the temperature–seroprevalence relationship is consistent and biologically plausible.

3) There is not much explanation why Tmin is in positive correlation with the force of infection.

4) Incorrect statement about average age at infection (page 9).

The sentence “Increasing the force of infection or the rate at which susceptible individuals become infected increases the average age of the first infection” is incorrect. In fact, the average age at first infection is inversely related to the force of infection: a higher force of infection typically leads to a lower average age at first infection. Please correct this sentence.

Reviewer #2: The objective of the study is clearly stated in the Introduction and the rationale for the models used is clearly explained. The population used for the modelling was clearly described and appropriate for the aim of the study.

Figures 2, 3 and S3 are cited in this section. This seems the best place for Figure 2 but I am not convinced it is the best place for Figure 3 or Figure S3. The information in Figure 3, especially Figures 3a and S3, has come from the literature search outlined in Figure S1. Consider including it in the results.

Results of the searches are also included in the text of the methods when they should be in the results section. There is also a discrepancy between the number cited in the text (87) and the number in Figure S1 (92).

Reviewer #3: In this article, the authors used data from a meta-analysis, supplemented it, and used other open-access data on demographic and climate projections to present what the burden of congenital toxoplasmosis in Africa could be in the future. The term “future” sound presumptuous, and a reformulation such as “based on demographic projections and scenarios of temperature change of plus or minus 3 degrees” would be more appropriate. The approach and reasoning are interesting, but the discussion of uncertainties and biases inherent in both meta-analysis and climate and demographic model projections should be further developed. For example, all meta-analyses document the means of data collection (sources and search strategies), summarize them, and discuss the value of the primary outcome measure in search of factors that could affect both the reproducibility and validity of the conclusions. The same rigor would be appropriate here and would help to alleviate some concerns:

- How was the search for additional articles conducted using the snowball method, and what elements made them relevant?

- What additional information did they provide compared to the meta-analysis by Gelaw et al.? Are they more or less heterogeneous? A table should represent and discuss them.

- What are the determinants and limitations of the open source upon demographic data? These elements should be considered in the discussion and general conclusion.

**Results**

-Does the analysis presented match the analysis plan?

-Are the results clearly and completely presented?

-Are the figures (Tables, Images) of sufficient quality for clarity?

Reviewer #1: The analysis follows the pre-specified plan.

However, there are several concerns regarding the results.

1) How was the temperature classification defined in Figure 3A?

2) The authors selected studies targeting pregnant women. Could the authors clarify why a wide range of age groups is included in Figure 3A?

3) It would be preferable to arrange the countries in the same order as in Figure 5 for consistency and easier comparison.

Reviewer #2: The analysis presented in the results matches the analysis outlined in the methods and the results are clearly presented. The figures are mostly of good quality although some of the information is hard to see but clarity could be improved. It might be due to me looking at a pdf but colours were hard to distinguish and the font size seemed small. More specific comments:

Figure 2: the first half of the figure is very well explained but there is almost nothing the figure legend about the second half. There is also very little about it in the text as it as the Figure is cited as part of the methods. It is an important point to understand so I would recommend adding a couple more sentences.

Figures 3 and S3: More information is needed in the legend. Make it clear that A is total seroprevalence. Explain why Ghana was chosen as the representative for fertility trajectory. Is there a reason why the seroprevalence in pregnant women is not included in the main paper? I think it would add value since the paper is about risks of congenital toxoplasmosis. Also, the colours in Fig S3 don't match the colours in Fig 3a but there is no explanation of what the colours relate to in the legend.

Reviewer #3: Tables and graphs allow results to be visualized and guide discussions. Care should be taken to ensure they are clear, and priority should be given to those that are most relevant. The results according to the different scenarios should be presented and discussed in greater detail and more clearly (conversely, Tables 1 and 2 could be provided as additional information). Figure 3 is only briefly discussed (just the part A). When it comes to temperature variations, shouldn't we pay more attention to the range than to minimum or maximum values? Figure 5 is an example of an overloaded graph: the presentation of these data should be revised to make them more readable (and discussable), for example by dividing them according to temperature scenario and demographic scenarios. A map-based presentation should also be considered.

**Conclusions**

-Are the conclusions supported by the data presented?

-Are the limitations of analysis clearly described?

-Do the authors discuss how these data can be helpful to advance our understanding of the topic under study?

-Is public health relevance addressed?

Reviewer #1: Authors clearly described the limitations of the study.

Reviewer #2: The conclusions are supported by the data presented and the authors have adequately addressed the limitations of the study. Even with the unavoidable limitations this study provides interesting results that will need to be considered by government authorities as they plan for a future where global temperatures rise.

Reviewer #3: Authors should formulate their results, comments, and conclusions with greater caution and reserve, partly because of biases and uncertainties (no work is entirely free of bias and as the X-Files series says, “the truth is out there”). On the other hand, care must be taken with the wording and the message that will be drawn from this work (we cannot suggest doing nothing to improve the HDI or the age of first pregnancy, so that the force of infection remains high and the risk of congenital toxoplasmosis remains low in the future). In addition, other hypotheses could be discussed by the authors: it is likely that temperature changes as significant as 3 degrees will cause radical changes to both ecosystems and economic systems and trigger significant population movements.

**Editorial and Data Presentation Modifications?**

Reviewer #1: (No Response)

Reviewer #2: I have mentioned a few issues with the figures in the results comments. They should be easy to address. Species names should be in italics.

Reviewer #3: Major revision, see comments above.

**Summary and General Comments**

Reviewer #1: (No Response)

Reviewer #2: This is a well-written paper that is relatively easy to follow. The authors have given very good explanations of why they chose the parameters they did and put the results into context of changes to infectious diseases transmission as the climate changes.

Reviewer #3: This work is interesting in that it brings together data on the seroprevalence of toxoplasmosis in Africa and demographic projections, and incorporates assumptions about temperature changes to estimate the force of infection by toxoplasmosis and the consequences for congenital toxoplasmosis. However, these are only estimates and probabilistic models, the respective limitations of which should be clarified and discussed. The initial data and treatment results should be presented more clearly and should provide more information for discussion.

The comparison between a virus (rubella virus) and a parasite (Toxoplasma gondii) should either be discussed in more detail (discussion of their differences, complexity of the parasitic cycle and its still unknown information content) or omitted. In addition, the reality of a vaccine against toxoplasmosis in human medicine deserves to be discussed, particularly in light of Africa's economic capabilities.

PLOS authors have the option to publish the peer review history of their article (what does this mean? ). If published, this will include your full peer review and any attached files.

**Do you want your identity to be public for this peer review?** For information about this choice, including consent withdrawal, please see our Privacy Policy .

Reviewer #1: **Yes:** Hyun Beom Song

Reviewer #2: No

Reviewer #3: No

**Figure resubmission:**
---

## [Decision Letter · Decision Letter 1]

8 Feb 2026

Evaluating how demography and temperature increase might alter the burden of congenital Toxoplasmosis in Africa

Dear Dr. Rasambainarivo,

Thank you for submitting your manuscript to PLOS Neglected Tropical Diseases. After careful consideration, we feel that it has merit but does not fully meet PLOS Neglected Tropical Diseases's publication criteria as it currently stands. Therefore, we invite you to submit a revised version of the manuscript that addresses the points raised during the review process.

* A letter that responds to each point raised by the editor and reviewer(s). You should upload this letter as a separate file labeled 'Response to Reviewers '. This file does not need to include responses to any formatting updates and technical items listed in the 'Journal Requirements' section below.

* A marked-up copy of your manuscript that highlights changes made to the original version. You should upload this as a separate file labeled 'Revised Manuscript with Track Changes '.

* An unmarked version of your revised paper without tracked changes. You should upload this as a separate file labeled 'Manuscript '.

We look forward to receiving your revised manuscript.

Kind regards,

Naomichi Yamamoto, Ph.D.

Academic Editor

Laura-Isobel McCall

Section Editor

Shaden Kamhawi

co-Editor-in-Chief

Paul Brindley

co-Editor-in-Chief

**Additional Editor Comments:**

All three reviewers have provided favorable comments on the manuscript. Please incorporate their comments to improve the manuscript. In particular, as Reviewer #3 pointed out, there is still room for improvement in the quality of the figures. Please take the reviewer's comments into consideration and work to improve the quality of the figures.

**Journal Requirements:**

**Reviewers' comments:**

Reviewer's Responses to Questions

**Key Review Criteria Required for Acceptance?**

**Methods**

-Are the objectives of the study clearly articulated with a clear testable hypothesis stated?

-Is the study design appropriate to address the stated objectives?

-Is the population clearly described and appropriate for the hypothesis being tested?

-Is the sample size sufficient to ensure adequate power to address the hypothesis being tested?

-Were correct statistical analysis used to support conclusions?

-Are there concerns about ethical or regulatory requirements being met?

Reviewer #1: Concerns were comprehensively addressed.

Reviewer #2: The authors have taken note and made most the suggested changes from myself and the other reviewers. I note that there are still results included in the methods section around the searches that were made to identify the seroprevalence data. This information should form the start of the Results section and I believe the associated figure (the PRISMA diagram) should be included as a proper figure rather than a supplementary one. It contains important data.

Reviewer #3: The objectives of this article are clearly stated and the design of this original study appears to be sound. The statistical analyses performed appear to be correct and support the conclusions stated.

**Results**

-Does the analysis presented match the analysis plan?

-Are the results clearly and completely presented?

-Are the figures (Tables, Images) of sufficient quality for clarity?

Reviewer #1: Concerns were comprehensively addressed, , though a few minor editorial details remain.

1) Figure 2: The Y-axis on the left plot is labeled "Prop serpositive by age 19," missing an 'o' in "seropositive".

2) Figure 3: The Y-axis label is misspelled as "Propoertion IgG Positive".

Reviewer #2: There have been significant improvements in how the results are presented. This section is much easier to follow now.

Reviewer #3: The analyses are as expected, but work remains to be done on the quality, clarity, and readability of the figures:

Figure 3 shows more colors than the legend, and the size of the dots suggests another parameter that is not shown. This creates a discrepancy between the figure and its presentation that needs to be corrected. Furthermore, the title of the vertical axis needs to be corrected.

Figure 4, which consists of several graphs, shows an overlap and should have a legend associated with it.

Figure 5 is also fragmented and of poor quality (the results for Eswatini are truncated, for example). The legend is unclear and the relationship between 2050 and 2100 should be explained in the text (why is there stagnation between 2050 and 2100 for Benin and a reversal of the relative burden between 2050 and 2100 for the countries shown below?). All documents placed in supplementary material should include a title and caption so that they can be understood independently of the text (in particular the Figure S5 where many colors and parameters are not explained).

**Conclusions**

-Are the conclusions supported by the data presented?

-Are the limitations of analysis clearly described?

-Do the authors discuss how these data can be helpful to advance our understanding of the topic under study?

-Is public health relevance addressed?

Reviewer #1: Concerns were comprehensively addressed.

Reviewer #2: The authors have responded well to the issues raised by the reviewers and this section is fine.

Reviewer #3: The conclusions are supported by the data and are written in a more objective manner, with greater attention paid to the limitations of the data and analyses, giving this study greater scope.

**Editorial and Data Presentation Modifications?**

Reviewer #1: (No Response)

Reviewer #2: I did notice some grammatical errors so it would be useful to get someone who hasn't viewed the paper frequently to proof-read it. They aren't major errors.

Reviewer #3: Minor revision : plase edit the figure and table.

**Summary and General Comments**

Reviewer #1: Concerns were comprehensively addressed, though a few minor editorial details remain as described above.

It is ultimately the authors' choice, but moving Figure S6 to the main text (perhaps as Figure 5, preceding the current Figure 5) would help demonstrate the separate effects of climate change and demographic shifts. This adjustment would allow for a more mechanistic understanding of how temperature affects risk, independent of demographic shifts, before presenting the total projected disease burden.

Reviewer #2: (No Response)

Reviewer #3: no comments

PLOS authors have the option to publish the peer review history of their article (what does this mean? ). If published, this will include your full peer review and any attached files.

**Do you want your identity to be public for this peer review?** For information about this choice, including consent withdrawal, please see our Privacy Policy .

Reviewer #1: **Yes:** Hyun Beom Song

Reviewer #2: No

Reviewer #3: No

**Figure resubmission:**
---

## [Editor Report · Decision Letter 2]

19 Feb 2026

Dear Rasambainarivo,

We are pleased to inform you that your manuscript 'Evaluating how demography and temperature increase might alter the burden of congenital Toxoplasmosis in Africa' has been provisionally accepted for publication in PLOS Neglected Tropical Diseases.

Best regards,

Naomichi Yamamoto, Ph.D.

Academic Editor

Laura-Isobel McCall

Section Editor

Shaden Kamhawi

co-Editor-in-Chief

Paul Brindley

co-Editor-in-Chief

The manuscript has been further improved by incorporating the reviewers' comments. I would like to thank the authors for their efforts. Regarding the PRISMA diagram, it is not required as it does not affect the content of the research, but I tend to support Reviewer #1's opinion that it should be included in the main text. The reason is that PLOS journals have no length restrictions, so it is fine to include it in the main text, and including it in the main text saves readers the time of having to refer to supplementary information. As an option, please consider including it in the main text, too.

---

## [Editor Report · Acceptance letter]

Dear Rasambainarivo,

We are delighted to inform you that your manuscript, "Evaluating how demography and temperature increase might alter the burden of congenital Toxoplasmosis in Africa," has been formally accepted for publication in PLOS Neglected Tropical Diseases.

Best regards,

Shaden Kamhawi

co-Editor-in-Chief

Paul Brindley

co-Editor-in-Chief
